# Self-assembly of metal–organic polyhedra into supramolecular polymers with intrinsic microporosity

Arnau Carné-Sánchez[1,4], Gavin A. Craig[1], Patrick Larpent[1], Takashi Hirose [2], Masakazu Higuchi[1], Susumu Kitagawa[1], Kenji Matsuda [2], Kenji Urayama [3] & Shuhei Furukawa [1,2]

Designed porosity in coordination materials often relies on highly ordered crystalline networks, which provide stability upon solvent removal. However, the requirement for crystallinity often impedes control of higher degrees of morphological versatility, or materials processing. Herein, we describe a supramolecular approach to the synthesis of amorphous polymer materials with controlled microporosity. The strategy entails the use of robust metal–organic polyhedra (MOPs) as porous monomers in the supramolecular polymerization reaction. Detailed analysis of the reaction mechanism of the MOPs with imidazole-based linkers revealed the polymerization to consist of three separate stages: nucleation, elongation, and cross-linking. By controlling the self-assembly pathways, we successfully tuned the resulting macroscopic form of the polymers, from spherical colloidal particles to colloidal gels with hierarchical porosity. The resulting materials display distinct microporous properties arising from the internal cavity of the MOPs. This synthetic approach could lead to the fabrication of soft, flexible materials with permanent porosity.

[1] Institute for Integrated Cell-Material Sciences (WPI-iCeMS), Kyoto University, Yoshida, Sakyo-ku, Kyoto 606-8501, Japan. [2] Department of Synthetic Chemistry and Biological Chemistry, Graduate School of Engineering, Kyoto University, Katsura, Nishikyo-ku, Kyoto 615-8510, Japan. [3] Department of Macromolecular Science and Engineering, Kyoto Institute of Technology, Matsugasaki, Sakyo-ku, Kyoto 606-8585, Japan. [4] Present address: Catalan Institute of Nanoscience and Nanotechnology (ICN2), CSIC, The Barcelona Institute of Science and Technology, Campus UAB, Bellaterra, 08193 Barcelona, Spain. Correspondence and requests for materials should be addressed to S.F. (email: shuhei.furukawa@icems.kyoto-u.ac.jp)

Nature's well-known aversion to empty spaces in condensed phases is explained by the tendency of molecules to maximize intermolecular interactions by packing densely[1]. Thus, the majority of porous materials known to date consist of long-range ordered extended networks that link their molecular or atomic building units through strong interactions (including ionic, covalent, and coordination bonds) to form structures with cavities[2–4]. The robust nature of the structural backbone enables evacuation of the cavities without structural collapse, leading to empty pores[5–7]. Meanwhile, a recent trend has revealed the potential of materials that are ordered on shorter length scales, where defects and disorder can lead to emergent properties such as softness and processability[8–12]. However, there is no clear strategy for preserving designed porosity in materials that do not present long-range order.

In contrast, porous organic cages (POCs) and metal–organic polyhedra (MOPs) are discrete molecules with predesigned porosity[13–15]. van der Waals interactions between the molecules dominate the packing arrangement, determining the overall stability in the solid state. The monomeric character of the molecules suggests that by their deliberate arrangement into soft materials, the need for order (whether short or long range) to enable the display of porosity can be overcome. Although approaches using porous molecules have been successfully applied to the synthesis of crystalline molecular solids[16–19], their applicability to the synthesis of amorphous materials with intrinsic and controlled porosity remains a challenge.

Herein, we outline a supramolecular approach to the formation of amorphous polymers with embedded porosity. Our design principle is to use a MOP as a porous monomer, and to polymerize them by taking advantage of the lability of metal–ligand coordination bonds. Comparable methodologies have been applied to obtain metallosupramolecular polymers for smart materials[20–22]; however, the predominant approach has consisted of linking single metal ions into the polymeric backbone, as the incorporation of large pre-organized molecular entities is more challenging. We use supramolecular polymerization to drive the concatenation of MOP monomers to incorporate porosity into macromolecular assemblies without the need for crystallinity. Advantageously the solubility of the MOP molecules allows us to follow the assembly process in solution and to understand the polymerization mechanism. As a consequence, we can choose the appropriate reaction pathway to obtain the desired macroscopic form of the polymers, whether spherical colloidal particles or colloidal gels, with the resulting porosity correlated to the macroscopic structure.

## Results and Discussion

**Design and reactivity of the porous monomers**. Our design for porous monomers requires soluble, stable, permanently porous MOPs (Fig. 1). However, the majority of reported MOPs rely on the Cu(II)–carboxylate bond, which is very labile and often causes structural decomposition upon desolvation and/or presents limited chemical stability[15]. We recently reported the synthesis of a highly stable cuboctahedral MOP based on a dirhodium paddlewheel motif, $[Rh_2(bdc)_2]_{12}$ (Fig. 2, $H_2bdc$ = benzene-1,3-dicarboxylic acid)[23]. The Rh–Rh single bond and strong Rh–carboxylate coordination bond confer high structural stability

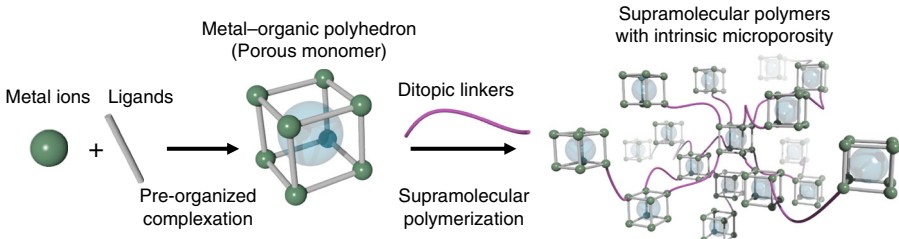

**Fig. 1** Supramolecular polymerization of porous monomers. Schematic illustration of the synthesis of MOPs as porous monomers and their subsequent coordination-driven self-assembly. Ditopic linkers are used to link the MOPs together to form supramolecular polymers with intrinsic porosity arising from the MOP units

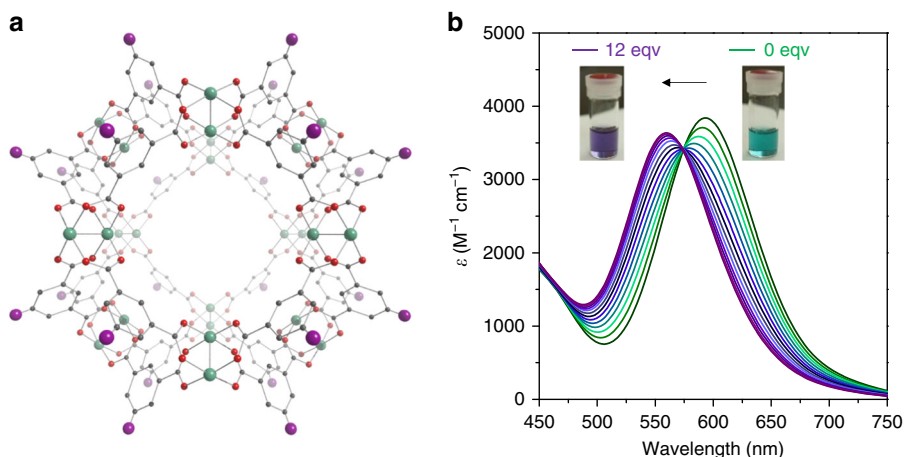

**Fig. 2** The $C_{12}RhMOP$ porous monomer. **a** Representation of the structure of the porous monomer $C_{12}RhMOP$ based on the crystal structure of the related MOP $[Rh_2(bdc)_2]_{12}$,[18] showing that it is formed by the coordination of rhodium ions (green) to the oxygen atoms (red) of the $H_2bdc$-$C_{12}$ ligand (gray, with the extended aliphatic chain simplified as a purple sphere). **b** UV–vis titration experiments of $C_{12}RhMOP$ (0.23 mM) with biz and (inset) a photo of the change in the color of the solution on progression from 0 mol. eq. (green) to 12 mol. eq. (purple)

on $[Rh_2(bdc)_2]_{12}$, and protect against collapse of the MOP upon desolvation and guest removal. In contrast to the equatorial sites of the dirhodium paddlewheel, the axial sites are highly labile, and are therefore ideal for polymerization through ligand exchange reactions with additional linkers. To ensure solubility of the porous monomer, we used an alkoxylated derivative of $H_2bdc$, yielding $[Rh_2(bdc-C_{12})_2]_{12}$ ($C_{12}RhMOP$, $H_2bdc-C_{12}$ = 5-dode-coxybenzene-1,3-dicarboxylic acid)[24].

For coordination-driven supramolecular polymerization we chose linkers with two imidazole functionalities because of the well-documented reactivity of the axial sites of dirhodium paddlewheels towards N-donor ligands[25–27]. Prior to polymerization experiments, the reactivity of $C_{12}RhMOP$ towards the monoimidazole ligand 1-benzylimidazole (biz) was studied. The Rh–Rh bond has a characteristic absorption band centered at 500–600 nm (band I), which corresponds to the $\pi^* \to \sigma^*$ transition and is highly sensitive to axial coordination, allowing the ligand exchange reaction of coordinated DMF for imidazole ligands to be followed spectroscopically[28, 29]. Control experiments with rhodium acetate, $[Rh_2(OAc)_4]$, showed that coordination of biz initially induces a shift of band I to $\lambda_{max}$ = 559 nm as the ligand binds to one of the available axial sites, while coordination to the second site drives a further observable shift to 540 nm (Supplementary Figure 1). In contrast, the coordination of 12 molar equivalents (mol. eq.) of biz to $C_{12}RhMOP$ induces a more limited incremental blue shift of the band I absorption maximum from 593 to 560 nm (Fig. 2b and Supplementary Figure 2). The presence of an isosbestic point at 574 nm throughout the titration experiments indicates that two possible coordination species are in equilibrium (one with biz bound to rhodium center and the other with DMF) and so each paddlewheel is a distinct chromophore, independent of the others in the MOP. Addition of more than 12 mol. eq. of biz did not induce any further shift in $\lambda_{max}$ (Supplementary Figure 3), and nuclear magnetic resonance (NMR) titration experiments indicated that the monoimidazole ligand coordinates solely to the exohedral axial sites of the paddlewheel unit (Supplementary Figure 4). Dynamic light scattering (DLS) measurements of the solution obtained after the addition of 12 mol. eq. of biz yielded a number-weighted size distribution of 3.2 ± 1.0 nm, consistent with isolated (i.e., non-aggregated) $C_{12}RhMOP$ molecules (Supplementary Figure 5). From these experiments, we concluded that the maximum coordination number for $C_{12}RhMOP$ is 12, with coordination of the imidazole ligands to the exterior, axial site of the paddlewheel.

**Formation of coordination polymer particles**. Following this calibration of the spectroscopic changes induced in the paddlewheel of the MOP by ligand exchange reactions in DMF, we then used this solvent medium for all subsequent polymerization reactions. To polymerize the $C_{12}RhMOP$ monomers, we used the ditopic imidazole linker 1,4-bis(imidazol-1-ylmethyl)benzene (bix, Fig. 3a). The stepwise addition of bix to a DMF solution of $C_{12}RhMOP$ was monitored through electronic absorption spectroscopy and DLS experiments. Addition of just 6 mol. eq. of bix to the solution induced a shift of $\lambda_{max}$ of band I from 593 to 558 nm, caused by coordination of both imidazole moieties of bix to all of the 12 available paddlewheel sites (Fig. 3b). The DLS experiments illustrate the assembly process of the $C_{12}RhMOP$ molecules into polymers as driven by the ligand exchange reaction: upon stepwise addition of bix (0.5 mol. eq. per step), the particle size grew from 3.2 ± 1.0 nm (isolated $C_{12}RhMOP$ molecules) to a maximum of 78 ± 5 nm after the addition of 6 mol. eq. (Supplementary Figure 6). Centrifugation of the resulting suspension allowed a purple powder to be isolated, which field-emission scanning electron microscopy (FESEM) showed to consist of particles having the characteristic spherical shape of

coordination polymer particles (CPP)[30, 31], with an average size of 70 ± 11 nm (CPP-1, Supplementary Figure 6). Infrared (IR) spectroscopy confirmed the presence of intact MOP units and also showed the characteristic bands of bix, while powder X-ray diffraction (PXRD) showed the particles to be amorphous (Supplementary Figures 7 and 8). The composition of CPP-1 was determined by $^1H$-NMR spectroscopy subsequent to acid digestion of the particles, and found to be $(C_{12}RhMOP)(bix)_6$ (MOP: bix = 1:6, Supplementary Figure 9). All of the bix molecules are incorporated in CPP-1 and act as bidentate ligands, leading to an observed $\lambda_{max}$ of band I at 560 nm in the diffuse reflectance UV–Vis spectrum (Supplementary Figure 10).

The temperature dependence of the growth of CPP-1 was investigated by variable-temperature DLS measurements (Fig. 3c). At 70 °C, the particles consist of isolated MOP molecules. Upon cooling, the particles remain isolated, until the assembly process is triggered at lower temperatures and a large increase in the particle size is observed. This temperature dependence of the growth is an indication that a nucleation–elongation mechanism underpins the assembly process, where $T_e$ is the temperature at which abrupt growth begins and is known as the elongation temperature[32]. $T_e$ was found to depend on the stoichiometric ratio of bix to a fixed concentration of $C_{12}RhMOP$ (Fig. 3c shows the experiments for $[C_{12}RhMOP]$ = 0.23 mM), with $T_e$ = 22, 34, and 50 °C for ratios of $C_{12}RhMOP$/bix of 1:0.5, 1:1, and 1:1.5, respectively. Increasing the total concentration of the components was observed to induce a further increase in $T_e$ (Supplementary Figure 11), which is consistent with recently reported nucleation-elongation-type supramolecular polymerization[33, 34]. In the absence of bix, no aggregation of $C_{12}RhMOP$ was observed (Supplementary Figure 12).

According to the nucleation and elongation model, separation of the nucleation step from the elongation process is key for control of the resulting particle size, in an analogous fashion to crystallization processes[35] and living supramolecular polymerizations[36, 37]. To investigate the supramolecular polymerization process of the $C_{12}RhMOP$ molecules triggered by the addition of the ditopic ligand, DLS experiments were used to observe the effect of the size of the steps employed in the stepwise addition of bix on the resulting size of the CPP-1 particles. Figure 3d shows the evolution of the size of CPP-1 during the addition of bix in steps of 0.25, 0.5, 1, or 2 molar equivalents until reaching a total of 6 added equivalents. Smaller steps led to bigger colloids: CPP-1 particles were obtained with sizes of 179 ± 11 nm (steps of 0.25 mol. eq.), 137 ± 6 nm (0.5 mol. eq.), 91 ± 12 nm (1 mol. eq.), and 51 ± 6 nm (2 mol. eq.), with the sizes observed using DLS consistent with FESEM images (Fig. 3e–g and Supplementary Figure 13). We propose that upon initial addition of fewer molar equivalents of bix, fewer nuclei form, resulting in larger colloids on finalizing the titration.

Assuming that all of the $C_{12}RhMOP$ molecules and 6 molar equivalents of bix present in the final solution are fully incorporated into CPPs, it follows that the size of CPP-1 should increase monotonically during the elongation process as a function of the volume of the added components. However, the DLS experiments show this not to be the case, and in fact the size of CPP-1 saturates when the molar equivalents of added bix reaches and exceeds 4 (Fig. 3d). UV–vis analyses of the supernatant solutions suggested that almost all of the MOP monomers are consumed in the assembly of the particles by this point (Supplementary Figure 14). Below 4 added mol. eq. of bix, the composition of the particles was inconsistent with the stoichiometry of the mixtures: the chemical compositions of CPP-1 particles after addition of 1 and 2 molar equivalents of bix were found to be $(C_{12}RhMOP)(bix)_{1.7}$ and $(C_{12}RhMOP)(bix)_3$, respectively (Supplementary Figures 15 and 16). Meanwhile, the

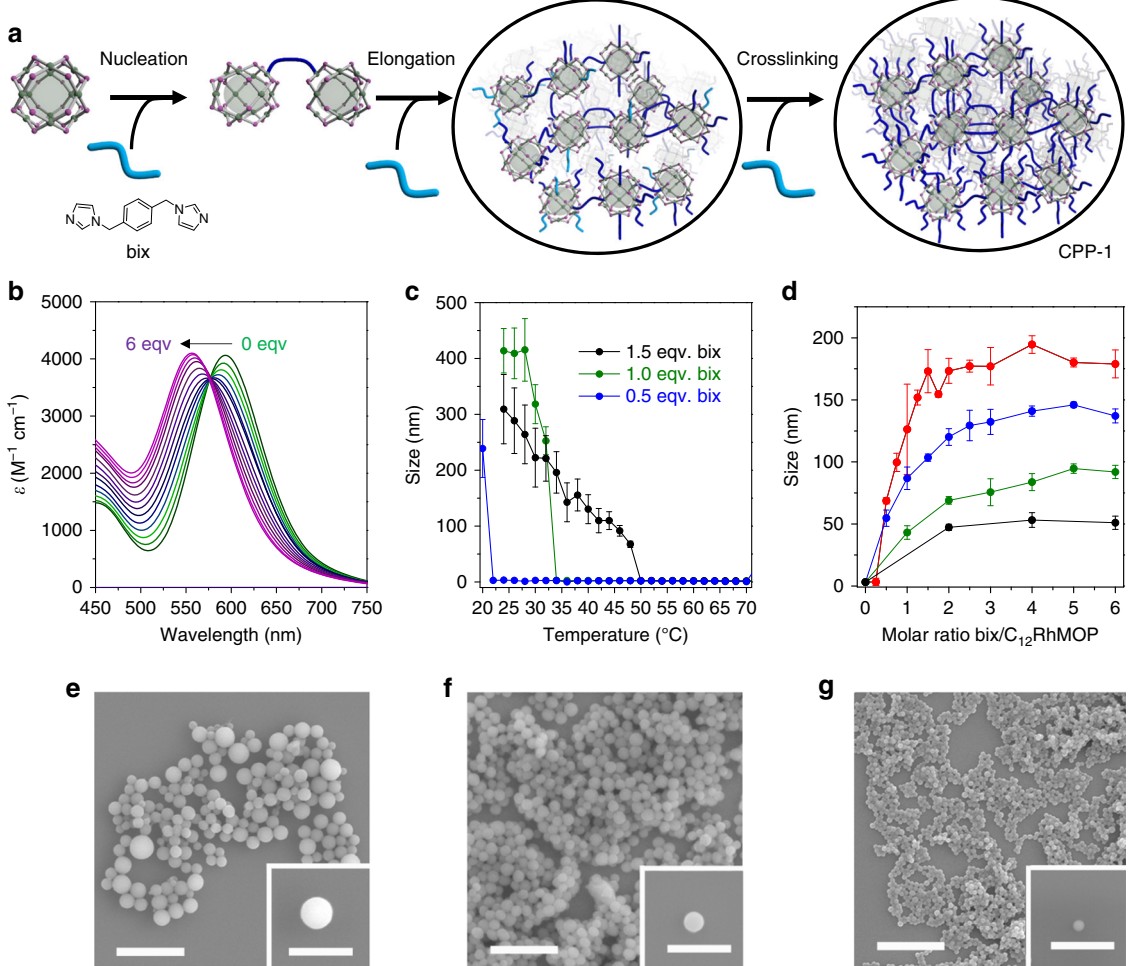

**Fig. 3** Stepwise polymerization of $C_{12}RhMOP$ leading to coordination polymer particles. **a** Schematic illustration of the reaction pathway of $C_{12}RhMOP$ with bix to yield CPP-1, which entails initial nucleation, followed by elongation upon addition of more bix molecules, and finally cross-linking, where the dark blue color of bix represents coordination of both imidazole rings. **b** UV–vis titration experiment showing the shift in band I of the rhodium paddlewheel in $C_{12}RhMOP$ that occurs during the formation of CPP-1, performed using an initial concentration of $C_{12}RhMOP$ of 0.23 mM. **c** DLS monitoring of the size evolution of CPP-1 nuclei (0.23 mM) synthesized after adding 0.5 (blue), 1 (green), and 1.5 mol. eq. (black) of bix at 80 °C and cooling to 20 °C. **d** Size evolution of the colloids obtained during the titration of $C_{12}RhMOP$ (0.93 mM) with bix as determined by DLS measurements. Titrations were performed by adding bix aliquots of 0.25 mol. eq. (red), 0.5 mol. eq. (blue), 1 mol. eq. (green), or 2 mol. eq. (black) at each step. Error bars correspond to the standard deviation of three repetitions. **e–g** FESEM images of CPP-1 synthesized by adding different amounts of bix (0.25 mol. eq. (**e**), 1 mol. eq. (**f**), and 2 mol. eq. (**g**) up to a molar ratio of bix/$C_{12}RhMOP$ = 6. Scale bars: 1 μm (inset = 500 nm)

compositions were $(C_{12}RhMOP)(bix)_4$ for addition of 4.0 mol. eq., and finally $(C_{12}RhMOP)(bix)_6$ (6.0 mol. eq.) (Supplementary Figures 17 and 9). Note that while the size of CPP-1 depends on the step size employed during addition, the chemical composition depends only on the total amount of bix added.

These results suggest the following nucleation–elongation–crosslinking mechanism for supramolecular polymerization of the MOP molecules: first, nucleation occurs upon initial addition of bix to the MOP solution, and nuclei of CPPs form; second, subsequent addition of bix increases the particle size via the supramolecular polymerization reaction with MOPs at the surface of the particles; and finally, size evolution stops when all of the MOP molecules in solution are consumed (here at MOP:bix = 1:4); with the remaining bix molecules diffusing into the colloids to crosslink between the MOPs, and yield a chemical composition of $(C_{12}RhMOP)(bix)_6$.

**Formation of supramolecular colloidal gels.** Recent work in supramolecular chemistry has shown that the nature of a

final assembly can depend on the degree of kinetic control of the reaction pathways[37–39]. One strategy to achieve kinetic control is to trap a metastable phase, before driving it to a more stable structure by applying a stimulus to overcome the required activation energy[40]. Thus, we envisaged that we would be able to obtain a metastable phase by forcing the coordinative saturation of discrete $C_{12}RhMOP$ molecules. To attain this control, an excess of bix molecules (12 mol. eq.) was added to the MOP solution at 80 °C followed by rapid cooling to room temperature, to form an isolated MOP molecule with the composition of $(C_{12}RhMOP)(bix)_{12}$, in which all of the bix molecules coordinate in a monodentate fashion (Fig. 4a, Supplementary Figure 18). The DLS analysis showed the presence of a stable isolated MOP with a size of 5.2 ± 1.2 nm and no sign of aggregation after 24 h at room temperature (Supplementary Figure 18).

In this situation, the driving force to trigger polymerization of this kinetically trapped $(C_{12}RhMOP)(bix)_{12}$ molecule cannot be enthalpy gain caused by the ligand exchange reaction, because all of the accessible axial sites of the paddlewheels are occupied by imidazole rings. The removal of bix from $(C_{12}RhMOP)(bix)_{12}$ can

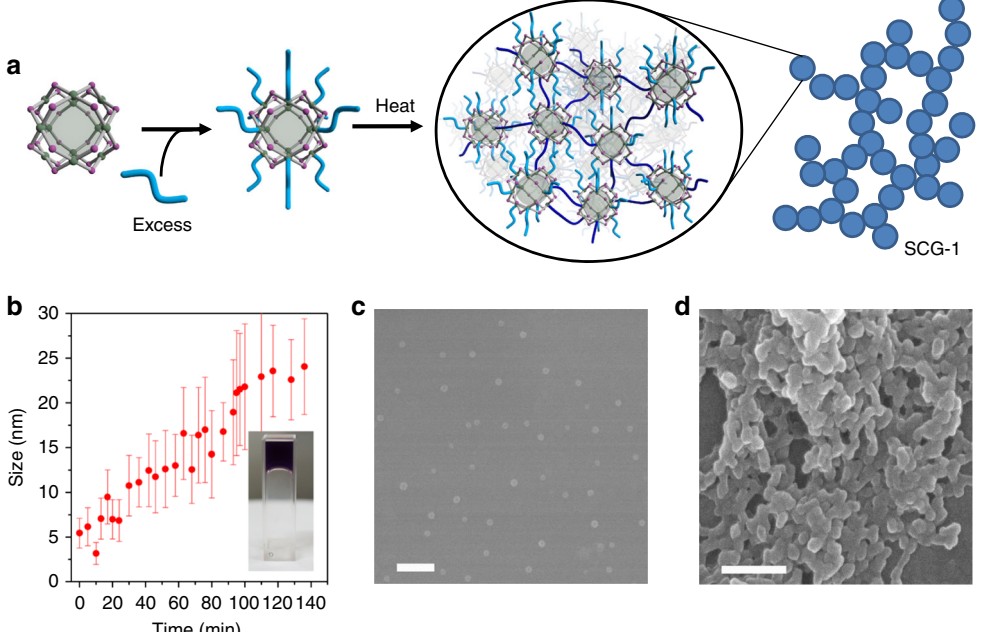

**Fig. 4** Polymerization of the kinetically trapped MOP ($C_{12}RhMOP$)(bix)$_{12}$ leading to the formation of a supramolecular colloidal gel. **a** Schematic representation of the proposed mechanism for the formation of SCG-1. **b** Size evolution of the kinetically trapped $C_{12}RhMOP$(bix)$_{12}$ (1.83 mM) molecules when heated at 80 °C, as followed with DLS measurements. (inset) A photo of SCG-1, as formed in the DLS cuvette during the experiment. **c** Isolated spherical particles obtained by quenching the SCG-1 formation reaction at 95 min. **d** Fused nanoparticles obtained by quenching the formation of SCG-1 upon reaching the plateau after 136 min. Scale bars: 200 nm

only induce linkage of MOPs as follows:

$$n(C_{12}RhMOP)(bix)_{12} \leftrightharpoons [(C_{12}RhMOP)(bix)_x]_n$$
$$+ n(12 - x)bix(6 < x < 12) \qquad (1)$$

where $[(C_{12}RhMOP)(bix)_x]_n$ represents supramolecular polymers. On the other hand, the release of free bix into the solution should result in an increase of entropy. Consequently, by heating the solution to 80 °C, it was possible to trigger the polymerization of ($C_{12}RhMOP$)(bix)$_{12}$ molecules into colloidal particles, measuring $22 \pm 7$ nm (Fig. 4b). Further incubation of these colloids led to the formation of a supramolecular colloidal gel (SCG-1, Fig. 4b inset and Supplementary Figure 19). SCG-1 was amorphous as evidenced by PXRD measurements (Supplementary Figure 20). Gel-like behavior was proven by rheological measurements, which showed the storage Young's modulus to be frequency independent ($E' \approx 10$ kPa) and one order of magnitude higher than the loss modulus (Supplementary Figure 21). Kinetic trapping of the ($C_{12}RhMOP$)(bix)$_{12}$ molecule is essential for the formation of the supramolecular gel: the sequential addition of bix to MOP up to 12 mol. eq. at 80 °C led to the formation of colloidal particles (Supplementary Figure 22).

To gain more insight into the gelation mechanism, two separate gelating mixtures were quenched by centrifugation, one prior to reaching the growth plateau (see Fig. 4b) at a size of $22 \pm 7$ nm (95 min) and the other after reaching the plateau (136 min), and the obtained solids were observed by FESEM. Before the plateau, the sample is composed of isolated spherical particles with an average size of $24 \pm 6$ nm, consistent with the size estimated by DLS measurements ($22 \pm 7$ nm, Fig. 4c). On the other hand, at the plateau and just before gelation, fused nanoparticles were observed (Fig. 4d). Therefore, the plausible gel formation mechanism proceeds through an initial stage of colloidal growth, followed by fusion of the colloids, leading to an extended coordination network that immobilizes solvents within[41].

The MOP unit withstood the gelation process, and the composition of SCG-1 was found to be ($C_{12}RhMOP$)(bix)$_{9.7}$ (MOP/bix = 1:9.7, Supplementary Figures 23 and 24). The excess of bix is understood by considering that not all of the bix molecules in SCG-1 act as bidentate linkers, but rather a fraction of them are anchored to the MOP as monodentate ligands. Taking into account that all of the dirhodium paddlewheels in $C_{12}RhMOP$ are coordinated by bix molecules (Supplementary Figure 25), the crosslinking degree (CLD) can be defined by the following equation:

$$CLD(\%) =$$
$$(\text{the number of bidentate bix})/(\text{the total amount of bix}) \times 100$$
$$(2)$$

For SCG-1, with a composition of ($C_{12}RhMOP$)(bix)$_{9.7}$, CLD is estimated to be 24%, while for CPP-1 with the composition of ($C_{12}RhMOP$)(bix)$_6$ CLD = 100%. Hence, the fusion of the colloidal nanoparticles observed in the formation of the SCG-1 can be attributed to the presence of reactive, monodentate bix ligands on the surface of the nanoparticles, which induce the fusing process for gelation.

The kinetic trapping strategy is versatile and was successfully used with other imidazole-based linkers with different backbone functionality, such as long flexible dodecyl aliphatic chains, 1,1'-(1,12-Dodecanediyl)bis[1H-imidazole] (bidod) and rigid biphenyl moieties, 4,4'-Bis(imidazol-1-ylmethyl)biphenyl (bibPh, Supplementary Figures 26 and 27). In both cases the reaction followed the pathway explained above, only differing in the size of the colloids prior to fusion ($247 \pm 48$ nm and $37 \pm 4$ nm for bidod and bibPh, respectively). This size difference strongly influences the final outcome: fused microparticles were obtained in the case of bidod (Supplementary Figure 28) and a supramolecular gel was obtained with bibPh (SCG-2, Supplementary Figure 29).

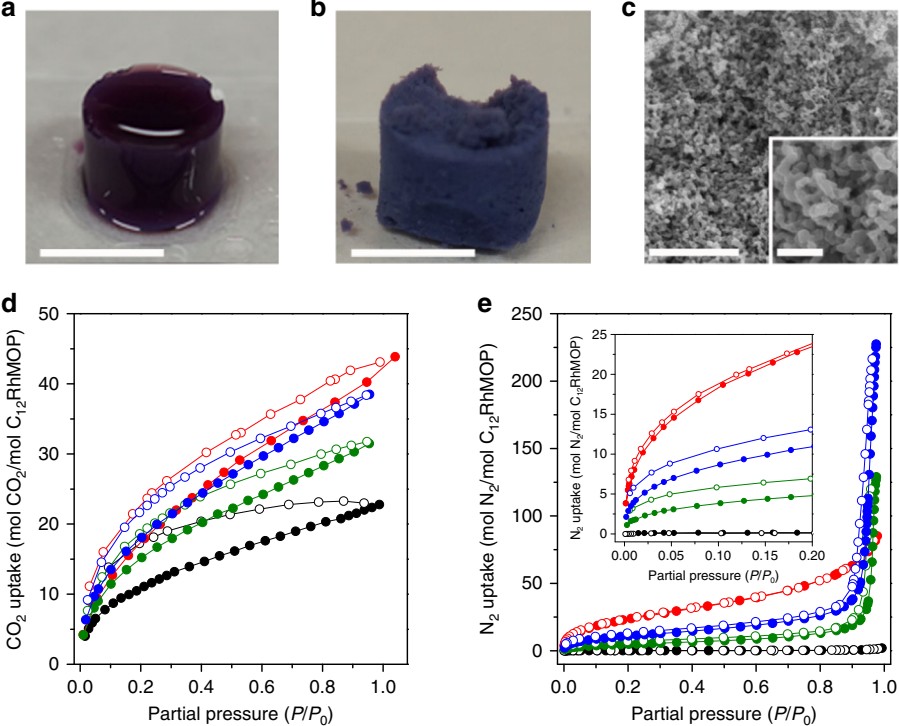

**Fig. 5** Drying of supramolecular colloidal gel leading to a supramolecular aerogel, and effect on gas sorption properties of the macroscopic structures of the polymers. **a** Photographs of a standalone SCG-1 (**a**) and **b**, the corresponding SAG-1; scale bars: 1 cm. **c** Representative FESEM image of SAG-1 and a magnified view of the material; scale bars: 1 μm, and (inset) 200 nm. **d** $CO_2$ adsorption isotherm at 195 K and **e** $N_2$ adsorption isotherm at 77 K of SAG-1 (red), CPP-1_small (blue), CPP-1_big (green), and $C_{12}RhMOP$ (black). The inset highlights the adsorption at low pressure. Filled and empty symbols correspond to adsorption and desorption, respectively

**Porosity of the supramolecular polymers**. While many crystalline MOFs are sufficiently rigid to avoid structural collapse after guest removal or activation and have been extensively studied in terms of porous properties[6], the porosity of amorphous coordination polymers is far less studied, perhaps due to their inability to withstand desolvation or a lack of structural data associated with the resulting amorphous phases. Although there are reports describing the synthesis of soft materials based on MOPs, confirmation of permanent porosity by gas adsorption experiments has never been demonstrated[42–45]. To prepare the supramolecular polymers for gas adsorption measurements, the materials were treated with supercritical $CO_2$, and finally activated by heating under vacuum (Supplementary Figure 30). The supercritical $CO_2$ drying process transformed SCG-1 into the corresponding aerogel SAG-1 (Fig. 5a, b; SAG = supramolecular aerogel). SAG-1 was found to consist of a hierarchical macroporous structure built up from the fused particles of an average size of $38 \pm 11$ nm (Fig. 5c).

The backbone of the supramolecular polymer presented here contains robust RhMOPs, which embed permanent microporosity in the amorphous network. The $CO_2$ adsorption experiments performed on CPP-1 and SAG-1 at 195 K demonstrate that the intrinsic porosity of the monomer is preserved through the supramolecular polymerization, as evidenced by the type 1 isotherm obtained in both cases (Fig. 5d). In addition, the $CO_2$ uptake at $P/P_0 = 0.95$ of SAG-1 (68.64 cm³/g and 40.23 mol $(CO_2)/mol(C_{12}RhMOP)$) clearly outperforms the porous monomer (46.01 cm³/g and 22.20 mol$(CO_2)/mol(C_{12}RhMOP)$, Supplementary Figure 31). This significant improvement can be ascribed to the nanostructure of SAG-1: the small particle size and the presence of macropores enhances gas diffusion and uptake, while $C_{12}RhMOP$ is a bulk powder containing aggregates in the

micrometric regime (Supplementary Figure 32). The importance of the nanoscale structure for gas sorption properties is further illustrated by comparing the gas uptake for two differently sized CPP-1 samples. The $CO_2$ uptake of CPP-1 particles of an average size of $53 \pm 10$ nm (CPP-1_small) was 38.50 mol$(CO_2)$/mol $(C_{12}RhMOP)$ (70.4 cm³/g), while the uptake of CPP-1 particles of an average size of $181 \pm 50$ nm (CPP-1_big) was 31.46 mol $(CO_2)/mol(C_{12}RhMOP)$ (57.53 cm³/g). As their molecular compositions are the same, the difference in $CO_2$ uptake can only be attributed to the difference in particle size.

The impact of the supramolecular polymer morphology on gas uptake was further investigated with $N_2$ adsorption experiments performed at 77 K. At low relative pressures, up to $P/P_0 \approx 0.1$, the discrete $C_{12}RhMOP$ displays a negligible adsorption of 0.17 mol $(N_2)/mol(C_{12}RhMOP)$. This performance is clearly improved upon by the supramolecular polymers: at this pressure, CPP-1_big, CPP-1_small, and SAG-1 present uptakes of $N_2$ measuring 3.82, 8.70, and 18.61 mol$(N_2)/mol(C_{12}RhMOP)$, respectively (Fig. 5e and Supplementary Figure 33). In addition, both CPP-1 samples show condensation of $N_2$ at high pressure ($P/P_0 \approx 0.9$), and so reach maximum uptakes of 128.35 and 227.44 mol$(N_2)$/ mol$(C_{12}RhMOP)$ for CPP-1_big and CPP-1_small, respectively— this type of condensation is typically observed in agglomerated nanoparticles due to interparticle voids. Meanwhile, the uptake displayed by SAG-1 resembles a type II isotherm, indicative of a wide distribution of pore sizes, consistent with the structure revealed by FESEM (Supplementary Figure 34).

In summary, we have demonstrated the supramolecular polymerization of Rh-based MOPs using imidazole linkers to form amorphous polymers with permanent porosity. By controlling the self-assembly pathway, we were able to selectively fabricate two distinct macroscopic morphologies: size-

controllable spherical particles, and three-dimensionally interconnected colloidal gels. The resulting amorphous materials showed gas adsorption properties arising from the microporosity of the MOP's internal cavity. The macroscopic morphology was found to strongly influence the adsorption properties of the assemblies. The strategy presented here is based on controlling the equilibrium of coordination bonds. Therefore, it can be conceivably generalized to any metal–organic polyhedron with appropriate coordination equilibria to linkers. While we have used a cuboctahedral MOP, we anticipate that porous monomers with other polyhedral geometries would offer another degree of control to attain different macroscopic structures. We envisage that by understanding the relationship between molecular scale geometries and resulting macroscopic shapes, then a real advance can be made towards the development of soft matter that is both permanently porous and amenable to materials processing.

## Methods

**Materials**. All chemicals reagents and solvents were purchased (Wako, Japan) and used without further purification. The detail protocols for ligand synthesis and the instrumentation can be found in Supplementary Methods.

**Synthesis of $C_{12}$RhMOP**. The synthesis of $C_{12}$RhMOP was adapted from previously reported procedures[24]. In a typical synthesis, rhodium acetate $Rh_2(A-cO)_4 \cdot (MeOH)_2$ (0.1 g, 0.2 mM) was reacted with 5-dodecoxybenzene-1,3-dicarboxylic acid ($H_2$BDC-$C_{12}$) (0.174 g, 0.5 mM) in 10 ml of N,N-dimethylacetamide (DMA) in the presence of $Na_2CO_3$ (0.052 g, 0.5 mM) at 100 °C for 48 h in a pre-heated oven. The resulting green solution was centrifuged and the supernatant treated with MeOH to precipitate the $C_{12}$RhMOP. The solid was redissolved in dichloromethane, filtered, and evaporated. Finally, the residue was redispersed and washed in ethanol and dried overnight at 120 °C under vacuum. Analysis calculated for $(C_{12}RhMOP)(H_2O)_6$ (FW = 10988.64 g/mol): C, 52.70%; H, 6.26 %. Found: C, 53.27%; H, 6.87 %.

**Synthesis of CPP-1**. In a typical experiment, a DMF solution of $C_{12}$RhMOP (0.92 mM) was titrated with a DMF solution of bix (12.5 mol. eq./ml) adding different molar equivalents until 6 mol. eq. The obtained purple suspension was centrifuged, washed with DMF, and then washed with acetone. The obtained particles were dried with supercritical $CO_2$ at 14 MPa and 40 °C for 90 min. Prior to sorption measurements, CPP-1 was activated at 120 °C under vacuum for 12 h. Analysis calculated for $(C_{12}RhMOP)(bix)_6$ (FW = 12310.38 g/mol): C, 55.25 %; H, 6.17 %; N, 2.74 %; Found: C, 54.04%; H, 6.14 %; N, 2.73 %.

**Synthesis of the kinetically trapped $C_{12}$RhMOP(bix)$_{12}$**. A hot DMF solution (80 °C) of $C_{12}$RhMOP was added to stirring DMF solution of 12 mol. eq. of bix to yield a final concentration of MOP of 0.23–1.93 mM. The purple solution was cooled to room temperature and used for further experiments.

**Synthesis of SCG-1 and SAG-1**. In a typical synthesis a solution of $C_{12}$RhMOP (bix)$_{12}$ (1.83 mM) was incubated at 80 °C overnight to induce gelation. The obtained gel was immersed in acetone for 3 days, replacing the solvent for fresh acetone each day. The solvent-exchanged gel was dried by supercritical $CO_2$ at 14 MPa and 40 °C for 90 min to obtain the SAG. Prior to sorption measurements, SAG-1 was activated at 120 °C under vacuum for 12 h. Analysis calculated for $(C_{12}RhMOP)(bix)_{9.7}$ (FW = 13192.05 g/mol): C, 56.27 %; H, 6.15 %; N, 4.13 %; Found: C, 56.85 %; H, 6.33 %; N, 5.03 %.

**Data availability**. The data that support the findings of this study are available from the corresponding author upon request.

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

## Acknowledgements

A.-C.S., G.A.C., P.L. are grateful to the JSPS Postdoctoral Fellowship Program for Foreign Researchers. We thank to Mr. Naoto Yamada (Kyoto Institute of Technology) for his help in performing rheological measurements and Dr. Kazunori Sugiyasu (National Institute for Materials Science) and Dr. Yasuhiro Ishida (RIKEN) for their fruitful discussions. This study was supported by JSPS KAKENHI Grant Number 15H03785 (Kiban B) and 17H05367 (Coordination Asymmetry). iCeMS is supported by the World Premier International Research Initiative (WPI), MEXT, Japan.

## Author contributions

A.-C.S., G.A.C., and S.F. conceived and designed the experiments. A.-C.S., G.A.C., and P. L. performed all synthetic and characterization experiments. A.-C.S., M.H., and S.K. performed sorption experiments. A.-C.S., G.A.C., T.H., and K.M. performed UV–Vis experiments. A.-C.S. and K.U. performed rheological measurements. A.-C.S., G.A.C., T. H., and S.F. analyzed the data and wrote the manuscript. All authors discussed the results and commented on the manuscript.

## Additional information

**Competing interests:** The authors declare no competing interests.

