## [Peer Review File · Nature Communications]

REVIEWERS' COMMENTS:

Reviewer #1 (Remarks to the Author):

The paper by Shuhei Furukawa and coworkers presents the detailed science of supramolecular polymerisation of metal-organic polyhedra into novel porous materials. I found reading this paper incredibly exciting as I feel it contributes to a number of research fields (supramolecular polymers, supramolecular gels, metal-organic polyhedra, molecular porous materials, materials science, nano science) and is a very well put together piece of science. I couldn't recommend publication more. And I sincerely hope that the reader community of Nature Communications will also embrace the paper.

The statistical appropriateness of the analytical data is good and I feel the science is highly reproducible.

Below are a few suggestions on increasing the quality of the paper and its readership.

Although it is clear to some degree from the experimental section that the solvent utilised in all the experiments is DMF, I would have liked the authors to state so in a part of the main text of the manuscript. e.g. All experiments of aggregation were performed in DMF.

I am not sure about the sentence "The presence of an isosbestic point at 574 nm throughout the titration experiments indicates that only two species are in equilibrium". The two species remark I feel is a little misleading. The spectrum could be indicative of two species, but the chemistry is 13 different compounds, not including any potential positional isomers. I would recommend rephrasing to indicate that the Rh centers behave independently of each other meaning the binding can be simplified to bound and unbound.

Due to my background I found that there are many possible connections with the literature of supramolecular gels to this work and to other porous materials. And that the paper contributes to the field of supramolecular gels significantly means I would have recommended publication with it being the only research field link, let alone the many that it does. There are a number of missed opportunities to link to this well established community. Metal binding supramolecular gels are well established materials with a number of reviews available to cite in this manuscript (Chem. Commun., 2016, 52, 8055-8074; Chem. Rev., 2010, 110, 1960-2004). Some of these metal based gels have been utilised for porous material development (Chem. Commun., 2017, 53, 8502-8505; Chem. Sci., 2017, 8, 3939-3948;).

An additional material class I believe should be highlighted here due to its similar design aspects to the paper are work by Stuart James and the concept of porous liquids (Nature, 2015, 527, 216-220).

One last paper I think should also be cited is from Foster and Steed titled "Exploiting Cavities in Supramolecular Gels" (Angew. Chem. Int. Ed., 2010, 49, 6718). The title I believe explains why I think it is important to acknowledge this paper and the work it cites.

Dr. Gareth O. Lloyd

Reviewer #2 (Remarks to the Author):

This is very interesting work that proposes and demonstrates a rather different approach to the construction of porous coordination solids than has been generally used so far. Specifically, instead of coordination polymerisation of metal ions and bridging ligands to generate crystalline infinite

frameworks that define pore space, prefabricated pores are used in the form of metal-organic polyhedra, and they are subsequently linked into networked structures through additional bridging ligands. The fact that the porosity is intrinsic to the MOPs means that the linking of these units into polymers does not need to lead to crystalline solids to generate porous materials, i.e. amorphous solids or gels can be produced which should retain porosity of the MOPs. The approach has the advantage of providing the solids in forms which have different processing characteristics to porous crystals. Overall I find the paper convincing and believe this is significant work that opens up new ways of making porous solids. I recommend publication after attention to the following points:

1. Abstract – first sentence – this does not appear to be true for coordination aerogels, please see <http://pubs.rsc.org.queens.ezp1.qub.ac.uk/en/content/articlepdf/2012/jm/c1jm14618a>

2. Intro first sentence – this holds true for solid and liquid states but for gases the avoidance of empty space is better explained by maximising entropy. I suggest qualifying this statement along the lines:

Nature's well-known aversion to empty spaces in condensed phases is explained by the tendency of molecules to maximise intermolecular interactions by packing densely.

3. P.3 "in materials of this kind." – For clarity, could the authors be explicit about the kind of materials they are referring to at this point please.

4. The use of the word 'elongation' in the mechanism is confusing to me since it implies 1-dimensional rather than 3-dimensional growth – could simply 'growth' be used instead?

5. P.13 : "the porosity of amorphous coordination polymers is far less studied due to their inability to withstand desolvation" – I'm not sure if this is justified, please see above regarding aerogels. Amorphous materials tend to be less studied I believe because of the lack of detailed structural characterisation rather than inherent instability to desolvation. I may be wrong, but the authors should reconsider this point I think and substantiate if needed.

6. I did not find exptl. details for UV-vis – which solvent was used for example? UV vis and NMR data point to coordination of the imidazole ligands and linkers only on the outside but there is no explanation offered for this. This behaviour is critical to the current work, but is also quite surprising, especially since the linkers would seem to be able to penetrate the MOPs on steric grounds. No explanation or discussion is given on this point. Do the authors have an explanation for this behaviour? Comments on this would be helpful.

7. "The resulting purple powder" – how was this obtained – some brief description should be given in the main body of the text.

8. Are the polymer spheres, i.e. the non-gelled materials, microporous?

9. "Gellification" should be gelation.

Reviewer #3 (Remarks to the Author):

This is an excellent paper that is at the interface of materials science and synthetic chemistry and I found the manuscript a pleasure to review. Furukawa and co-workers have shown that supramolecular polymers constructed from MOP units can maintain permanent porosity in the solid-state. Furthermore, they demonstrate a deep understanding of how these materials form and control of their morphology. I believe this paper is a significant advance in the area of supra-

molecular chemistry (and in my opinion may well lead to a renaissance in MOP chemistry) and is of a scientific standard that meets the requirements of Nature Commun. It is obvious the authors have taken great care and effort in preparing all aspects of this manuscript. The paper is written in clear and concise language and the characterisation is performed in a scholarly manner and from this perspective I don't believe significant revisions are necessary. However, I think the paper could be strengthened by revisiting the section on porosity. Ascertaining the origin of porosity in amorphous materials is challenging. In this case the authors refer to the intrinsic porosity of the MOP being preserved and giving rise, or at least contributing significantly, to permanent porosity. This may be, and is likely true, however the N₂ and CO₂ suggest that other mechanisms may also be at play and do not provide conclusive evidence that the origin of porosity arises from the MOPs. I think experimental pore-size distributions would strengthen this argument. The N₂ isotherms are Type II which in addition to the authors comments can be observed for materials with a higher external surface area with respect to internal surface area, also the CO₂ isotherms do not necessarily prove significant contributions from the intrinsic pores of the MOPs. I don't believe extra experiments are required as this would be enough work for a follow-up study, however, perhaps the discussion around porosity could be 'tightened' a little prior to publication.

Response to reviewer #1

Comments:

The paper by Shuhei Furukawa and coworkers presents the detailed science of supramolecular polymerisation of metal-organic polyhedra into novel porous materials. I found reading this paper incredibly exciting as I feel it contributes to a number of research fields (supramolecular polymers, supramolecular gels, metal-organic polyhedra, molecular porous materials, materials science, nano science) and is a very well put together piece of science. I couldn't recommend publication more. And I sincerely hope that the reader community of Nature Communications will also embrace the paper.

The statistical appropriateness of the analytical data is good and I feel the science is highly reproducible.

Below are a few suggestions on increasing the quality of the paper and its readership.

Comment 1: Although it is clear to some degree from the experimental section that the solvent utilised in all the experiments is DMF, I would have liked the authors to state so in a part of the main text of the manuscript. e.g. All experiments of aggregation were performed in DMF.

Response:

The start of the section “Formation of coordination polymer particles” on p6 has now been re-written to read:

“Following this calibration of the spectroscopic changes induced in the paddlewheel of the MOP by ligand exchange reactions in DMF, we then used this solvent medium for all subsequent polymerization reactions.”

Comment 2: I am not sure about the sentence “The presence of an isosbestic point at 574 nm throughout the titration experiments indicates that only two species are in equilibrium”. The two species remark I feel is a little misleading. The spectrum could be indicative of two species, but the chemistry is 13 different compounds, not including any potential positional isomers. I would recommend rephrasing to indicate that the Rh centers behave independently of each other meaning the binding can be simplified to bound and unbound.

Response:

We appreciate the indication of the reviewer and we modified the text in the manuscript according to his suggestion as follows:

Page 6, line 7: “The presence of an isosbestic point at 574 nm throughout the titration experiments indicates that two possible coordination species are in equilibrium (one with biz

bound to rhodium center and the other with DMF) and so each paddlewheel is a distinct chromophore, independent of the others in the MOP”

Comment 3: Due to my background I found that there are many possible connections with the literature of supramolecular gels to this work and to other porous materials. And that the paper contributes to the field of supramolecular gels significantly means I would have recommended publication with it being the only research field link, let alone the many that it does. There are a number of missed opportunities to link to this well established community. Metal binding supramolecular gels are well established materials with a number of reviews available to cite in this manuscript (Chem. Commun., 2016, 52, 8055-8074; Chem. Rev., 2010, 110, 1960–2004). Some of these metal based gels have been utilised for porous material development (Chem. Commun., 2017, 53, 8502-8505; Chem. Sci., 2017, 8, 3939–3948;). An additional material class I believe should be highlighted here due to its similar design aspects to the paper are work by Stuart James and the concept of porous liquids (Nature, 2015, 527, 216–220). One last paper I think should also be cited is from Foster and Steed titled "Exploiting Cavities in Supramolecular Gels" (Angew. Chem. Int. Ed., 2010, 49, 6718). The title I believe explains why I think it is important to acknowledge this paper and the work it cites.

Response:

We agree with Reviewer #1 about the necessity of including further references acknowledging the previous work done in the fields of supramolecular gels [Chem. Commun., 2016, 52, 8055-8074; Chem. Rev., 2010, 110, 1960–2004], the utilization of cavities in such materials [Angew. Chem. Int. Ed., 2010, 49, 6718], MOF-based gels and aerogels [Chem. Commun., 2017, 53, 8502-8505; Chem. Sci., 2017, 8, 3939–3948;] and novel porous materials such the porous liquids [Nature, 2015, 527, 216–220]. Therefore, all suggested references have been included in the main text as follows:

Chem. Commun., 2017, 53, 8502-8505; Chem. Sci., 2017, 8, 3939–3948 and Nature, 2015, 527, 216–220 have referenced in *page 3, line 10* as references 10-12.

Chem. Commun., 2016, 52, 8055-8074 and Chem. Rev., 2010, 110, 1960–2004 have been cited in *page 10, line 9* as references 39 and 38 respectively.

Angew. Chem. Int. Ed., 2010, 49, 6718 is cited in *page 13, line 12* as reference 42.

Response to reviewer #2

Comments:

This is very interesting work that proposes and demonstrates a rather different approach to the construction of porous coordination solids than has been generally used so far. Specifically, instead of coordination polymerisation of metal ions and bridging ligands to generate crystalline infinite frameworks that define pore space, prefabricated pores are used in the form of metal-organic polyhedra, and they are subsequently linked into networked structures through additional bridging ligands. The fact that the porosity is intrinsic to the MOPs means that the linking of these units into polymers does not need to lead to crystalline solids to generate porous materials, i.e. amorphous solids or gels can be produced which should retain porosity of the MOPs. The approach has the advantage of providing the solids in forms which have different processing characteristics to porous crystals. Overall I find the paper convincing and believe this is significant work that opens up new ways of making porous solids. I recommend publication after attention to the following points:

Comment 1: Abstract – first sentence – this does not appear to be true for coordination aerogels, please see <http://pubs.rsc.org.queens.ezp1.qub.ac.uk/en/content/articlepdf/2012/jm/c1jm14618a>

Response:

We thank Reviewer #2 for bringing this paper to our attention. However, in the abstract we intended to highlight the fact that the porosity that arise in amorphous materials cannot be controlled or “designed” as in the case of crystalline coordination polymers (such as MOFs). This is because in the amorphous coordination gels it is difficult to predict the coordination number of the metal node, the presence of higher nuclearity metal clusters or the coordination mode of the linkers. Thus, even though some amorphous supramolecular gels have been shown to be porous, there is no general strategy to control the environment of the pore, its size and the apertures to access it. However, in order to acknowledge the fact that porosity can also be displayed by amorphous coordination polymers we rephrased the first sentence of the abstract as follows:

page 2, line 1: “Designed porosity in coordination materials often relies on highly ordered crystalline networks, which provide stability upon solvent removal.”

Comment 2: Intro first sentence – this holds true for solid and liquid states but for gases the avoidance of empty space is better explained by maximising entropy. I suggest qualifying the statement along the lines:

Response:

We rephrased the sentence as suggested by reviewer #2

page 3, line 2: “Nature’s well-known aversion to empty spaces in condensed phases is explained by the tendency of molecules to maximise intermolecular interactions by packing densely.”

Comment 3: P.3 “in materials of this kind.” – For clarity, could the authors be explicit about the kind of materials they are referring to at this point please.

Response:

In order to clarify our statement we rephrase the sentence as follows:

page 3, line 10: “However, there is no clear strategy for preserving designed porosity in materials that do not present long-range order.”

Comment 4: The use of the word ‘elongation’ in the mechanism is confusing to me since it implies 1-dimensional rather than 3-dimensional growth – could simply ‘growth’ be used instead?

Response:

In the field of supramolecular polymerization, and more precisely molecular self-assembly, the term elongation is used when the polymerization is best described by the cooperative model rather than by the isodesmic model. Thus, “elongation” is not related to the morphology of the final product but to the mechanism followed to synthesise it. In fact, the word elongation has been previously used to describe the formation of two- and three-dimensional supramolecular structures [Nat. Chem., 2017, 9, 493-499]. In our case, the word elongation seems suitable as the growth of CPP-1 is better described by the nucleation-elongation model.

Comment 5: P.13 : “the porosity of amorphous coordination polymers is far less studied due to their inability to withstand desolvation” – I’m not sure if this is justified, please see above regarding aerogels. Amorphous materials tend to be less studied I believe because of the lack of detailed structural characterisation rather than inherent instability to desolvation. I may be wrong, but the authors should reconsider this point I think and substantiate if needed.

Response:

We thank the referee for this alternative suggestion. We have re-written the sentence to include this possibility:

page 13, line 2 : “the porosity of amorphous coordination polymers is far less studied, perhaps due to their inability to withstand desolvation or a lack of structural data associated with the resulting amorphous phases.”

Comment 6: I did not find exptl. details for UV-vis – which solvent was used for example? UV vis and NMR data point to coordination of the imidazole ligands and linkers only on the outside but there is no explanation offered for this. This behaviour is critical to the current work, but is also quite surprising, especially since the linkers would seem to be able to penetrate the MOPs on steric grounds. No explanation or discussion is given on this point. Do the authors have an explanation for this behaviour? Comments on this would be helpful.

Response:

All experiments have been carried out in DMF and the text has been modified to include this clarification at the start of the section “Formation of coordination polymer particles” as follows;

page 6: “Following this calibration of the spectroscopic changes induced in the paddlewheel of the MOP by ligand exchange reactions in DMF, we then used this solvent medium for all subsequent polymerization reactions.”

Regarding the coordination of imidazole linkers, it is important to highlight that the C₁₂RhMOP has two different apertures, which, are wider on the exohedral side than in the inner cavity due to the linker orientation [Chem, 2017, 2, 393-403]. Thus, the calculated aperture of the square and triangular window in solution is 0.58 nm and 0.39 nm respectively. This small apertures prevent the ligands from entering the inner cavity as demonstrated by UV-VIS and NMR experiments.

Comment 7: “The resulting purple powder” – how was this obtained – some brief description should be given in the main body of the text.

Response:

The powder was obtained by centrifugation, and we have re-written this sentence to make it more explicit:

page 7, line 6: “Centrifugation of the resulting suspension allowed a purple powder to be isolated, which field-emission scanning electron microscopy (FESEM) showed to consist of particles having the characteristic spherical shape of coordination polymer particles (CPP)”

Comment 8: Are the polymer spheres, i.e. the non-gelled materials, microporous?

Response:

As shown in Figure 5d, the spherical coordination polymeric particles also present a type 1 CO₂ isotherm showing the presence of microporosity.

Comment 9: "Gellification" should be gelation.

Response:

This has been changed accordingly.

Response to reviewer #3

Comments:

This is an excellent paper that is at the interface of materials science and synthetic chemistry and I found the manuscript a pleasure to review. Furukawa and co-workers have shown that supramolecular polymers constructed from MOP units can maintain permanent porosity in the solid-state. Furthermore, they demonstrate a deep understanding of how these materials form and control of their morphology. I believe this paper is a significant advance in the area of supra-molecular chemistry (and in my opinion may well lead to a renaissance in MOP chemistry) and is of a scientific standard that meets the requirements of Nature Commun. It is obvious the authors have taken great care and effort in preparing all aspects of this manuscript. The paper is written in clear and concise language and the characterisation is performed in a scholarly manner and from this perspective I don't believe significant revisions are necessary.

Comment 1: However, I think the paper could be strengthened by revisiting the section on porosity. Ascertaining the origin of porosity in amorphous materials is challenging. In this case the authors refer to the intrinsic porosity of the MOP being preserved and giving rise, or at least contributing significantly, to permanent porosity. This may be, and is likely true, however the N₂ and CO₂ suggest that other mechanisms may also be at play and do not provide conclusive evidence that the origin of porosity arises from the MOPs. I think experimental pore-size distributions would strengthen this argument. The N₂ isotherms are Type II which in addition to the authors comments can be observed for materials with a higher external surface area with respect to internal surface area, also the CO₂ isotherms do not necessarily prove significant contributions from the intrinsic pores of the MOPs. I don't believe extra experiments are required as this would be enough work for a follow-up study, however, perhaps the discussion around porosity could be 'tightened' a little prior to publication.

Response:

We thank reviewer #3 for raising this important issue. The use of the heavily functionalized C₁₂RhMOP with long alkoxy chains as a porous monomer was motivated by its solubility, which enabled the study of its self-assembly in solution, but it came with the cost of limiting the porosity. Thus, even though its CO₂ adsorption isotherm at 195 K suggest that most of its porosity comes from the internal cavity this could not be corroborated by a pore size distribution (PSD) analysis due to its lack of adsorption of N₂ at 77K. Therefore, it is plausible that the observed porosity of C₁₂RhMOP is the result of a combination of the contribution of the internal cavity and, to some extent, the extrinsic porosity. The same applies to the supramolecular polymers, which also present a type 1 CO₂ adsorption isotherm at 195 K. In addition, both SAG-1 and CPP-1_small seem to have a higher contribution of the microporosity judging from their higher N₂ uptake at lower partial pressure probably due to the structuring into nanoparticles or colloidal gels. The future work will develop the characterization strategy to fully clarify this issue.

Finally, to make the discussion of the porous properties of the C₁₂RhMOP and its related supramolecular polymers more precise we modified the third paragraph of page 14 as follows:

Page 14, line 9: “The impact of the supramolecular polymer morphology on gas uptake was further investigated with N₂ adsorption experiments performed at 77 K. At low relative pressures, up to $P/P_0 \approx 0.1$, the discrete C₁₂RhMOP displays a negligible adsorption of 0.17 mol(N₂)/mol(C₁₂RhMOP). This performance is clearly improved upon by the supramolecular polymers: at this pressure, CPP-1_big, CPP-1_small, and SAG-1 present uptakes of N₂ measuring 3.82, 8.70, and 18.61 mol(N₂)/mol(C₁₂RhMOP), respectively (Fig. 5e and Supplementary Figure 33). In addition, both CPP-1 samples show condensation of N₂ at high pressure ($P/P_0 \approx 0.9$), and so reach maximum uptakes of 128.35 and 227.44 mol(N₂)/mol(C₁₂RhMOP) for CPP-1_big and CPP-1_small, respectively – this type of condensation is typically observed in agglomerated nanoparticles due to interparticle voids. Meanwhile, the uptake displayed by SAG-1 resembles a type II isotherm, indicative of a wide distribution of pore sizes, consistent with the structure revealed by FESEM (Supplementary Figure 34).”